# Modeling in Forestry Using Mixture Models Fitted to Grouped and Ungrouped Data

Eric K. Zenner [1,*] and Mahdi Teimouri [2]

1   Department of Ecosystem Science and Management, The Pennsylvania State University, 305 Forest Resources Building, University Park, PA 16802, USA
2   Department of Statistics, Faculty of Science and Engineering, Gonbad Kavous University, No. 163, Basirat Blvd, Gonbad Kavous 4971799151, Iran; teimouri@aut.ac.ir
*   Correspondence: eric.zenner@psu.edu; Tel.: +1-814-865-4574

**Abstract:** The creation and maintenance of complex forest structures has become an important forestry objective. Complex forest structures, often expressed in multimodal shapes of tree size/diameter (DBH) distributions, are challenging to model. Mixture probability density functions of two- or three-component gamma, log-normal, and Weibull mixture models offer a solution and can additionally provide insights into forest dynamics. Model parameters can be efficiently estimated with the maximum likelihood (ML) approach using iterative methods such as the Newton-Raphson (NR) algorithm. However, the NR algorithm is sensitive to the choice of initial values and does not always converge. As an alternative, we explored the use of the iterative expectation-maximization (EM) algorithm for estimating parameters of the aforementioned mixture models because it always converges to ML estimators. Since forestry data frequently occur both in grouped (classified) and ungrouped (raw) forms, the EM algorithm was applied to explore the goodness-of-fit of the gamma, log-normal, and Weibull mixture distributions in three sample plots that exhibited irregular, multimodal, highly skewed, and heavy-tailed DBH distributions where some size classes were empty. The EM-based goodness-of-fit was further compared against a nonparametric kernel-based density estimation (NK) model and the recently popularized gamma-shaped mixture (GSM) models using the ungrouped data. In this example application, the EM algorithm provided well-fitting two- or three-component mixture models for all three model families. The number of components of the best-fitting models differed among the three sample plots (but not among model families) and the mixture models of the log-normal and gamma families provided a better fit than the Weibull distribution for grouped and ungrouped data. For ungrouped data, both log-normal and gamma mixture distributions outperformed the GSM model and, with the exception of the multimodal diameter distribution, also the NK model. The EM algorithm appears to be a promising tool for modeling complex forest structures.

**Keywords:** complexity; diameter (DBH) distribution; estimation-maximization (EM) algorithm; forest structure; gamma; log-normal; maximum-likelihood (ML) method; nonparametric kernel-density estimation; Weibull



## 1. Introduction

In forestry, the contemporary paradigms of ecological forestry and close-to-nature silviculture look toward natural disturbance regimes to inform management approaches [1,2]. In general, natural disturbances and silvicultural cutting practices are important drivers of forest dynamics that often cause partial upper canopy tree mortality and create openings that serve as sites and niches for the establishment of new and/or the release of advance tree regeneration [3–7]. Due to tremendous variabilities in size, intensity, severity, and frequency of disturbances [4,8], large live legacy trees often survive natural disturbances and are thus placed in the immediate proximity of the new regeneration [7,9–11]. The outcomes of such partial disturbances are highly heterogeneous vertical forest structures in time and

space that range from single to two or more cohorts or single to two or multiple canopy layers formed by pure or mixed tree species [12].

To quantify vertical forest structures, model forest dynamics, growth, and yield, and compare managed to natural stands, size/diameter at breast height (DBH) distributions are typically used [13]. Several shapes of DBH distributions have been consistently found in both managed and natural stands [14]. For example, relatively even-aged stands are often characterized by unimodal and near normal distributions, uneven-aged stands typically exhibit rotated sigmoid or reverse-*J* diameter distributions and multi-aged stands such as many old-growth forests show multimodal and/or irregularly descending distributions characterized by asymmetry, skewness, interruptions (i.e., gaps in the distribution), multimodality, and heavy tails [14].

Two general approaches (i.e., nonparametric and parametric methods) dominate the modeling of DBH distributions. The nonparametric kernel-based density estimation (NK) method e.g., Ref. [15] is a very efficient approach with the capacity to approximate a wide variety of shapes, but suffers from two restrictions: (i) the need for a suitable choice of bin size [16,17] and (ii) the inability to estimate standard errors and confidence intervals for the estimated parameters. Whereas NK methods smooth empirical DBH distributions without the need to estimate parameters, parametric methods estimate the parameters of particular probability density functions that can conform to a wide array of DBH distributions and are credited with providing a deeper understanding of forest dynamics [18]. Parametric methods are often differentiated into flexible single or one-component parametric probability density functions (PDFs) [19–22] and finite mixture models consisting of two or more components [23–26]. Recent research has shown that despite their flexibility in shape, popular single/one-component parametric models such as the gamma and Weibull PDFs do not adequately portray irregular bi- and multi-modal DBH distributions [15,25]. It is precisely these bi- and multi-modal DBH distributions that represent heavy-tailed and highly skewed DBH distributions with some large trees and gaps in the distribution, however, that reflect outcomes of natural disturbances that are of interest to forest scientists and managers [15,25]. For these latter, more complex distributions, finite mixture distributions that contain a few more components and thus define different shape parameters for the sub-populations that comprise the overall DBH distribution, are now recommended [27,28].

Finite mixture distributions treat overall, heterogeneous DBH distributions as a compound of several distributions of sub-populations composed of multiple basic shapes that reflect different age cohorts or canopy layers [15,18,29]. Among the recent studies that have investigated various mixture distributions, we refer to [18] for two-component mixture Weibull distributions, [30] for two-component mixtures of three-parameter Weibull distributions, [29] for the mixture of log-normal, normal and two-parameter gamma distributions, [23] for the mixture of two-parameter Weibull distributions, [31] for mixture of normal, log-normal, and two-parameter Weibull distributions, [25] for the mixture of three-parameter gamma and Weibull distributions, [15] for the mixture of two- and three-parameter gamma and Weibull distributions, [24] for the mixture of three-parameter Weibull distributions, and [14,28] for two-component mixtures of two-parameter gamma distributions and gamma-shaped mixture (GSM) models. The GSM model is a special type of the two-parameter gamma mixture model for which estimating the parameters is much easier, because the scale parameters of all components in each GSM model are the same and the shape parameter of the k-th component is k, for $k = 1, \ldots, K$ [27,32]. Hence, for a K-component GSM model 2K parameters are estimated whereas for a K-component gamma mixture model the number of parameters is 3K-1.

The parameters of the finite mixture models (including the GSM model) can be estimated via a Bayesian or maximum likelihood (ML) approach e.g., [14,15,25,28]. The Bayesian approach uses Gibbs samplers or Markov chain Monte Carlo (MCMC) computations that are computationally cumbersome and time consuming for large K. The ML approach requires suitable starting parameters and iteratively executes and completes com-

putations when numerical algorithms converge to a global maximum or terminates once a convergence criterion is met. Iterative algorithms such as the expectation-maximization (EM) are often combined with the Newton-Raphson (NR) method to find suitable initial values to estimate the parameters of the distributions [33]. Finding a suitable choice of the initial values for implementing the ML approach is not an easy task, however. Because the number of initial values that must be chosen depends on the number of groups (i.e., diameter classes) and the numbers of the components of the finite mixture model, this task becomes particularly challenging when the number of components for which starting values must be found is more than three or four. The NR method is, however, sensitive to the initial values, and the choice of initial values can strongly influence the speed of convergence of the estimation procedure and its ability to locate the global maximum [25]. Finally, complex forests structures characterized by irregular and multimodal empirical DBH distributions often exhibit many local maxima, minima, and saddle points on the likelihood surface that can cause algorithms to become extremely unstable and to not converge on the global maximum. In contrast, the iterative EM algorithm always converges, and does so expeditiously when a good starting value is chosen [34]. The K-means clustering algorithm is a powerful method for obtaining initial values [35] that can then be used in the iterative EM algorithm for estimating the parameters of a mixture model [33].

Although recent research on finite mixture models has predominantly focused on the gamma and Weibull PDFs, e.g., [14,15,25,28], less attention has been given to the log-normal mixture model, which is also a very flexible PDF that is able to fit highly skewed and heavy-tailed DBH distributions. Further, we have found no research devoted to estimating the parameters of finite mixture models fitted to grouped data in the field of forestry.

In this study, we expand on previous work on finite mixture models in forestry and developed the EM algorithm for the most commonly used multi-component mixture models in forestry, i.e., the gamma, Weibull, and log-normal, mixture models for conditions when observations are available both in grouped and in ungrouped form. In this study, we used the K-means clustering algorithm to obtain initial values for the EM parameter estimation approach. The objective of this paper is to explore the utility of the EM algorithm for fitting gamma, log-normal, and Weibull mixture models to diameter distributions of three irregular and multi-sized/aged forests. We further compared the performance of the resulting models with the best fit to the NK and GSM models when DBH data were ungrouped. Finally, we explored whether the number of groups (i.e., the number and width of diameter classes) influenced which model family provided the better fit in our empirical example datasets.

## 2. Results

The EM algorithm resulted in mixture models that fit the empirical data quite well. The model family that produced the closest fit with lowest Akaike Information Criterion (AIC) and largest log likelihood (LL) values differed among the three samples, with the log-normal and gamma families typically producing better fits than the Weibull family. The number of components in the various K-component mixture models that resulted in the best model when data were grouped depended on the width of the diameter classes and, to a minor extent, on the evaluation criterion (AIC or LL). Whether data were analyzed in grouped or ungrouped form did not influence the selection of the best model family and number of model components. For reference, the initial values for implementing the EM algorithm for the three sample plots obtained using the k-means clustering approach are given in Appendix C.

Sample 1 was characterized by a broadly bimodal, rotated sigmoid DBH distribution that had no trees in the size classes between 30 and 50 cm in DBH (Figure A1). The DBH distribution was best captured by two- or three-component models (Table 1). The two-component mixture model was the superior model for grouped-5 and ungrouped (based on AIC), whereas the three-component model was superior for grouped-2.5, grouped-5, and ungrouped (the latter two based on LL). The shapes fit by the two-component mixture

model families were similar, with the log-normal and gamma families providing the better fit (Figure A1a–d). In general, all models underestimated the density of trees in the smallest size class, particularly in grouped-2.5, and successfully reflected the bimodal shape of the underlying DBH distribution. The shapes of the curves of grouped-2.5 and grouped-5 as well as of grouped and their corresponding ungrouped data forms were very similar. Among the three-component mixture models, the log-normal family was consistently identified as superior for grouped and ungrouped data (Table 1). The three-component mixture models identified three modes in the underlying DBH distribution, but the different model families disagreed in the placement of the modes, particularly when the data were in grouped form (Figure A1e–h). For grouped-2.5, the log-normal distinguished between the smallest DBH class (first mode) and the 17.5–27.5 cm classes (second mode) and identified the 62.5–65 cm classes as representing the third mode of trees in the 52.5–87.5 cm size classes. In contrast, the gamma and Weibull families identified a single mode between 12.5–27.5 cm but distinguished two modes in the 52.5–87.5 cm size classes. For ungrouped data, the three model families provided similar shapes of the fitted curves and agreed on placing two modes in the 12.5–27.5 cm size classes. For grouped-5, the three model families agreed more closely on the fitted shape, with similar shapes of the log-normal and gamma families for size classes up to 32.5 cm and similar shapes of the gamma and Weibull families for the size classes above 52.5 cm. Despite some differences, the shapes of fitted curves of grouped and ungrouped data were generally similar.

**Table 1.** Parameter estimates and goodness-of-fit statistics for the mixture of log-normal, gamma, and Weibull distributions fitted to sample 1 when DBH data are ungrouped (UG), grouped in classes of width 2.5 cm (G2.5), and grouped in classes of width 5 cm (G5) . It should be noted that the estimated vector of mixing parameters is not given for the sake of saving space.

| | | | **Estimated Parameters** | | **Statistic** | |
|---|---|---|---|---|---|---|
| **K** | **Type** | **Family** | $\alpha = (\alpha_1, \ldots, \alpha_K)'$ | $\beta = (\beta_1, \ldots, \beta_K)'$ | **AIC** | **LL** |
| 2 | G2.5 | log-normal | $(4.210, 2.894)'$ | $(0.153, 0.288)'$ | 73.476 | $-31.738$ |
| | | gamma | $(11.848, 40.816)'$ | $(1.592, 1.671)'$ | 74.028 | $-32.014$ |
| | | Weibull | $(3.553, 6.460)'$ | $(20.971, 73.501)'$ | 76.675 | $-33.337$ |
| | G5 | log-normal | $(4.216, 2.890)'$ | $(0.266, 0.148)'$ | 45.312 | $-17.656$ |
| | | gamma | $(13.399, 44.042)'$ | $(1.395, 1.556)'$ | 46.154 | $-18.077$ |
| | | Weibull | $(3.487, 6.855)'$ | $(20.796, 73.218)'$ | 49.115 | $-19.557$ |
| | UG | log-normal | $(2.876, 4.213)'$ | $(0.287, 0.157)'$ | 207.174 | $-98.587$ |
| | | gamma | $(39.902, 12.080)'$ | $(1.714, 1.532)'$ | 207.815 | $-98.907$ |
| | | Weibull | $(6.397, 3.629)'$ | $(73.265, 20.551)'$ | 211.038 | $-100.519$ |
| 3 | G2.5 | log-normal | $(2.600, 3.108, 4.203)'$ | $(0.0371, 0.1722, 0.153)'$ | 65.557 | $-24.778$ |
| | | gamma | $(334.213, 108.835, 12.499)'$ | $(0.251, 0.575, 1.512)'$ | 70.185 | $-27.092$ |
| | | Weibull | $(22.398, 12.600, 3.724)'$ | $(86.140, 64.990, 20.979)'$ | 70.969 | $-27.484$ |
| | G5 | log-normal | $(4.110, 4.135, 2.947)'$ | $(0.023, 0.104, 0.253)'$ | 46.263 | $-15.131$ |
| | | gamma | $(15.288, 90.470, 0.122)'$ | $(1.288, 0.694, 0.122)'$ | 46.670 | $-15.335$ |
| | | Weibull | $(45.341, 10.354, 3.836)'$ | $(83.511, 63.075, 21.076)'$ | 48.099 | $-16.049$ |
| | UG | log-normal | $(4.441, 4.136, 2.876)'$ | $(0.049, 0.094, 0.287)'$ | 210.274 | $-97.137$ |
| | | gamma | $(416.582, 112.260, 12.080)'$ | $(0.204, 0.560, 1.532)'$ | 210.673 | $-97.336$ |
| | | Weibull | $(8.048, 7.846, 6.397)'$ | $(25.679, 15.343, 73.265)'$ | 214.205 | $-99.102$ |

Sample 2 was characterized by a broadly bimodal to multimodal distribution with an understory cohort between 12.5–45 cm in DBH, no trees between 45–65 cm in DBH, and a small overstory cohort between 65–75 cm in DBH (Figure A2). The understory cohort had a mode in the 22–22.5 cm class and one in the 30–35 cm classes. Overall, the two-component Weibull was the superior model for grouped-5 and ungrouped whereas the three-parameter gamma and log-normal were superior for grouped-2.5 (Table 2). The shapes fit by the two-component mixture model families were similar, particularly of the log-normal and

gamma families, but the Weibull family provided the better fit by placing the first mode correctly between 30–35 cm and providing a closer fit to the second mode between 65–75 cm (Figure A2a–d). In general, all models underestimated the density of trees in the first in grouped-2.5, but successfully reflected the bimodal shape of the underlying DBH distribution. The shapes of the curves of grouped-2.5 and grouped-5 as well as of grouped and their corresponding ungrouped data forms were very similar. Among the three-component mixture models, the log-normal and gamma families were consistently identified as superior for grouped and ungrouped data (Table 2). This is largely because of nearly identical shapes of the three-component log-normal and gamma mixture models that identified the same modes, whereas the Weibull family missed the actual mode of the underlying DBH distribution, particularly for grouped-5 (Figure A2e–h). Even though the first mode was a more obvious feature of grouped-2.5, both log-normal and gamma models portrayed this mode more sharply for grouped-5 but overestimated the density of the 30–35 cm and 65–75 cm classes. All three model families captured the empty size classes between 45–60 cm. For ungrouped data, the three model families provided similar shapes of the fitted curves and agreed rather well on placing the modes, with slightly better fits provided by the log-normal and gamma than the Weibull model. The overall relative advantage of the log-normal and gamma over the Weibull mixture distribution was a consistent similar shape for grouped and ungrouped forms.

**Table 2.** Parameter estimates and goodness-of-fit statistics for the mixture of log-normal, gamma, and Weibull distributions fitted to sample 2 when DBH data are ungrouped (UG), grouped in classes of width 2.5 cm (G2.5), and grouped in classes of width 5 cm (G5) . It should be noted that the estimated vector of mixing parameters is not given for the sake of saving space.

| | | | Estimated Parameters | | Statistic | |
|---|---|---|---|---|---|---|
| K | Type | Family | $\alpha = (\alpha_1, \ldots, \alpha_K)'$ | $\beta = (\beta_1, \ldots, \beta_K)'$ | AIC | LL |
| 2 | G2.5 | log-normal | $(4.230, 3.318)'$ | $(0.025, 0.289)'$ | 72.305 | $-31.152$ |
| | | gamma | $(972.111, 13.375)'$ | $(0.071, 2.142)'$ | 69.215 | $-29.607$ |
| | | Weibull | $(52.200, 4.698)'$ | $(69.976, 31.530)'$ | 63.122 | $-26.561$ |
| | G5 | log-normal | $(4.203, 3.322)'$ | $(0.028, 0.276)'$ | 41.739 | $-15.869$ |
| | | gamma | $(278.370, 14.189)'$ | $(0.240, 2.022)'$ | 40.463 | $-15.232$ |
| | | Weibull | $(38.627, 4.643)'$ | $(68.320, 31.567)'$ | 35.990 | $-12.995$ |
| | UG | log-normal | $(4.234, 3.318)'$ | $(0.037, 0.289)'$ | 304.256 | $-147.128$ |
| | | gamma | $(732.216, 13.388)'$ | $(0.094, 2.143)'$ | 300.953 | $-145.476$ |
| | | Weibull | $(39.990, 4.706)'$ | $(70.189, 31.421)'$ | 294.357 | $-142.178$ |
| 3 | G2.5 | log-normal | $(4.223, 3.018, 3.481)'$ | $(0.032, 0.240, 0.133)'$ | 58.660 | $-21.330$ |
| | | gamma | $(638.393, 18.451, 59.230)'$ | $(0.106, 1.128, 0.553)'$ | 58.522 | $-21.261$ |
| | | Weibull | $(41.376, 4.926, 7.611)'$ | $(69.697, 21.784, 35.645)'$ | 60.877 | $-22.438$ |
| | G5 | log-normal | $(4.200, 2.943, 3.469)'$ | $(0.020, 0.168, 0.135)'$ | 38.490 | $-11.245$ |
| | | gamma | $(632.997, 37.412, 53.350)'$ | $(0.105, 0.507, 0.605)'$ | 38.658 | $-11.329$ |
| | | Weibull | $(38.628, 6.351, 7.438)'$ | $(68.320, 21.144, 35.285)'$ | 42.186 | $-13.0932$ |
| | UG | log-normal | $(4.234, 2.972, 3.477)'$ | $(0.037, 0.228, 0.123)'$ | 300.263 | $-142.131$ |
| | | gamma | $(732.216, 20.710, 65.696)'$ | $(0.094, 0.967, 0.496)'$ | 299.940 | $-141.970$ |
| | | Weibull | $(39.990, 5.932, 8.117)'$ | $(70.189, 21.657, 35.461)'$ | 300.493 | $-142.246$ |

Sample 3 was characterized by a generally negative exponential or reverse-*J* diameter distribution typical of many old-growth forests where the number of trees initially declines sharply with increasing tree size (Figures A3 and A4). In contrast to the previous two samples, the diameter distribution in sample 3 was much wider, contained very few empty diameter classes toward the larger end of the long right-tailed distribution (between 80 and 100 cm DBH; Figures A3 and A4), and was mildly multi-modal. The multi-modality was largely smoothed over by all two-component mixture models but were more clearly expressed in the three- and four-component mixture models. Overall, the three-component

log-normal was the superior model for grouped-2.5 and grouped-5 (based on AIC) whereas the four-parameter log-normal was superior for grouped-2.5 and grouped-2.5 (based on LL) and ungrouped (Tables 3 and 4). Though broadly similar, the shapes of the three- and four-component log-normal and gamma models differed from the Weibull model at the first mode around the 10 cm size class, for which the Weibull provided the better fit, particularly for grouped-5, and around the mid-sized classes between 20–40 cm, which the Weibull model typically underestimated. For ungrouped data, the three model families provided similar shapes of the fitted curves and agreed rather well on placing the modes, with slightly better fits provided by the log-normal and gamma than the Weibull model. All three mixture model families provided curves that fit the data much more closely when the data were analyzed in ungrouped form.

**Table 3.** Parameter estimates and goodness-of-fit statistics for the mixture of log-normal, gamma, and Weibull distributions fitted to sample 3 when DBH data are ungrouped (UG) and grouped (G) in classes of width 2.5 cm. It should be noted that the estimated vector of mixing parameters is not given for the sake of saving space.

| | | | **Estimated Parameters** | | **Statistics** | |
|---|---|---|---|---|---|---|
| **K** | **Type** | **Family** | $\alpha = (\alpha_1, \ldots, \alpha_K)'$ | $\beta = (\beta_1, \ldots, \beta_K)'$ | **AIC** | **LL** |
| 1 | G | log-normal | 2.956 | 0.693 | 214.360 | −105.180 |
| | | gamma | 2.046 | 12.233 | 259.537 | −127.768 |
| | | Weibull | 1.407 | 35.634 | 329.422 | −162.7114 |
| | UG | log-normal | 2.954 | 0.693 | 3201.910 | −1598.955 |
| | | gamma | 2.045 | 12.223 | 3272.889 | −1634.444 |
| | | Weibull | 1.368 | 27.634 | 3307.548 | −1651.774 |
| 2 | G | log-normal | $(4.000, 2.668)'$ | $(0.350, 0.435)'$ | 175.664 | −82.832 |
| | | gamma | $(7.531, 5.853)'$ | $(7.269, 2.606)'$ | 181.498 | −85.749 |
| | | Weibull | $(3.421, 2.377)'$ | $(70.085, 18.450)'$ | 202.398 | −96.199 |
| | UG | log-normal | $(4.101, 2.689)'$ | $(0.263, 0.448)'$ | 3140.958 | −1565.479 |
| | | gamma | $(14.141, 5.081)'$ | $(4.403, 3.207)'$ | 3159.619 | −1574.517 |
| | | Weibull | $(3.796, 2.308)'$ | $(67.822, 18.481)'$ | 3194.808 | −1592.176 |
| 3 | G | log-normal | $(4.165, 2.376, 3.062)'$ | $(0.248, 0.268, 0.423)'$ | 174.497 | −79.248 |
| | | gamma | $(12.858, 13.141, 12.418)'$ | $(4.797, 1.857, 0.955)'$ | 176.244 | −80.122 |
| | | Weibull | $(4.704, 2.496, 3.875)'$ | $(75.592, 31.643, 13.306)'$ | 193.224 | −88.612 |
| | UG | log-normal | $(4.238, 3.520, 2.556)'$ | $(0.189, 0.228, 0.352)'$ | 3139.964 | −1561.982 |
| | | gamma | $(27.672, 18.870, 8.462)'$ | $(2.551, 1.840, 1.610)'$ | 3152.290 | −1566.975 |
| | | Weibull | $(5.448, 4.466, 3.152)'$ | $(77.088, 38.036, 15.169)'$ | 3188.805 | −1586.149 |
| 4 | G | log-normal | $(4.331, 3.922, 3.146, 2.428)'$ | $(0.167, 0.218, 0.306, 0.288)'$ | 180.238 | −79.119 |
| | | gamma | $(40.550, 29.524, 21.736, 12.860)'$ | $(1.860, 1.583, 1.124, 0.935)'$ | 184.617 | −81.308 |
| | | Weibull | $(7.234, 4.908, 3.170, 3.418)'$ | $(87.458, 59.458, 32.177, 14.167)'$ | 199.350 | −88.675 |
| | UG | log-normal | $(4.329, 3.892, 3.236, 2.462)'$ | $(0.143, 0.144, 0.188, 0.292)'$ | 3138.762 | −1558.381 |
| | | gamma | $(47.339, 48.153, 28.019, 12.138)'$ | $(1.621, 1.028, 0.924, 1.007)'$ | 3142.216 | −1560.108 |
| | | Weibull | $(7.656, 6.784, 5.427, 4.000)'$ | $(52.661, 82.269, 28.017, 13.269)'$ | 3172.867 | −1575.433 |

In all three samples, the log-normal mixture model generally did a comparable, if not better, job fitting the ungrouped data than the gamma mixture model and both models outperformed the GSM (Table 5). The log-normal mixture model was identified as superior to the gamma mixture model by all three goodness-of-fit measures in sample 1 and the Kolmogorov-Smirnov (KS) and the Cramér-von Mises (CVM) measures in sample 2, and the KS and Anderson-Darling (AD) measures in sample 3 (indicated by boldface values in Table 5). The NK model was identified as the superior model by the KS measure in sample 2 and by all three measures in sample 3 while the GSM model consistently exhibited the worst performance in all three samples. The differences in goodness-of-fit among the three

model families can be readily seen when the PDFs of the log-normal mixture, NK, and GSM models were superimposed onto the DBH distributions of all three samples (Figure A5).

**Table 4.** Parameter estimates and goodness-of-fit statistics for the mixture of log-normal, gamma, and Weibull distributions fitted to sample 3 in grouped case when class width is 5 cm. It should be noted that the estimated vector of mixing parameters is not given for the sake of saving space.

| | | Estimated Parameters | | Statistics | |
|---|---|---|---|---|---|
| K | Model | $\alpha = (\alpha_1, \ldots, \alpha_K)'$ | $\beta = (\beta_1, \ldots, \beta_K)'$ | AIC | LL |
| | log-normal | 2.950 | 0.700 | 136.568 | −66.284 |
| 1 | gamma | 2.017 | 12.376 | 179.552 | −87.776 |
| | Weibull | 1.403 | 34.300 | 237.321 | −116.660 |
| | log-normal | $(3.531, 2.481)'$ | $(0.568, 0.303)'$ | 103.765 | −46.882 |
| 2 | gamma | $(3.259, 9.486)'$ | $(12.567, 1.366)'$ | 107.470 | −48.735 |
| | Weibull | $(3.638, 2.276)'$ | $(70.807, 18.911)'$ | 132.321 | −61.160 |
| | log-normal | $(4.145, 3.104, 2.379)'$ | $(0.257, 0.368, 0.198)'$ | 95.901 | −39.950 |
| 3 | gamma | $(9.607, 12.329, 19.674)'$ | $(6.160, 1.843, 0.569)'$ | 96.210 | −40.105 |
| | Weibull | $(6.104, 2.586, 2.626)'$ | $(83.387, 48.790, 17.167)'$ | 129.290 | −56.645 |
| | log-normal | $(4.320, 3.881, 3.098, 2.388)'$ | $(0.170, 0.227, 0.296, 0.200)'$ | 101.564 | −39.782 |
| 4 | gamma | $(46.628, 23.178, 18.438, 16.192)'$ | $(0.935, 3.0233, 1.298, 0.716)'$ | 103.357 | −40.678 |
| | Weibull | $(18.360, 9.149, 2.697, 2.630)'$ | $(95.022, 71.635, 46.513, 17.165)'$ | 132.513 | −55.256 |

**Table 5.** Goodness-of-fit statistics for fitting mixture of gamma, mixture of log-normal, GSM, and NK models to samples 1 and 2 in ungrouped case. The number of components used for fitting the log-normal and gamma mixture models to samples 1 and 2 are 2 and 3, respectively. The number of components used for fitting GSM model to both samples is 250.

| | | Family | | | |
|---|---|---|---|---|---|
| Sample | Measure | Gamma | Log-Normal | NK | GSM |
| | KS | 0.090 | **0.088** | 0.143 | 0.106 |
| 1 | AD | 0.214 | **0.198** | 0.457 | 0.399 |
| | CVM | 0.024 | **0.022** | 0.056 | 0.045 |
| | KS | 0.110 | **0.104** | 0.100 | 0.128 |
| 2 | AD | **0.218** | 0.221 | 0.255 | 0.671 |
| | CVM | 0.035 | **0.034** | 0.046 | 0.124 |
| | KS | 0.039 | **0.039** | 0.026 | 0.094 |
| 3 | AD | 0.519 | **0.475** | 0.328 | 4.397 |
| | CVM | 0.069 | 0.074 | 0.021 | 0.436 |

## 3. Discussion

The EM algorithm was quite successful fitting two- and three-parameter gamma, log-normal, and Weibull mixture distributions to three empirically observed example diameter distributions. Empirical diameter distributions in natural forests are often characterized by random, local, irregular, and multiple modes that reflect peaks of establishment of natural regeneration at certain time intervals and/or episodic growth releases of individual or small groups of trees following disturbances or gap dynamics [4,14,36]. Theoretical probability density functions anticipate gradual, but not necessarily small, differences in the frequency of trees in neighboring size classes and are generally more successful at approximating multimodal than irregular distributions [25]. Thus, as long as empirical diameter distributions do not exhibit large, erratic differences in the frequency of trees in neighboring size classes (i.e., irregular distributions), kernel density and theoretical density functions can provide smoothed fits that closely approximate various shapes of empirical diameter distributions. In this study, flexible two- and three-parameter log-normal and gamma mixture models were particularly successful at approximated two differently

shaped, highly skewed, heavy tailed, and multimodal empirical DBH distributions with empty size classes.

The log-normal, gamma, and Weibull mixture distribution families are often used to analyze heterogeneous lifetime or survival data [37], which DBH distributions represent. In this study, these three mixture model distribution families did not perform equally well, however. In most cases, but not always, the log-normal and the gamma mixture models provided a closer fit to the empirical diameter distributions than the Weibull model; in most cases, the log-normal mixture models were only slightly superior to the gamma mixture models. Although mixture models are able to approximate stands with multimodal diameter distributions with high accuracy and fit DBH distributions very well around the largest maxima [14,15,29], the observed qualitative differences among the distribution families were largely due to an underestimation of the tree frequency in the class of the global maximum density, with more gradual and delayed changes in the shape of the Weibull mixture distribution than the log-normal and gamma mixture distributions (Figures A1 and A2).

The accuracy of the fit of all three mixture models also depended on the number of components of each model. AIC and LL indicated that two components generated the superior model in sample 1 and three or four components provided the better fit for the DBH distributions in samples 2 and 3 , reflecting the fact that the number of components in mixture models should be related to the number of maxima observed in a distribution [14]. Whereas all mixture models using fewer components than the number of maxima smoothed over and missed some local maxima and generally overestimated densities in empty classes, mixture models that matched the number of maxima were able to more closely trace the subtleties of the DBH shapes. As [15] point out, however, the choice of using a two- or three-component model depends largely on the study objective. If the study objective is to fit theoretical distribution models as precisely as possible to a specific empirical data set, then the number of mixture components should be matched to the number of random, local extremes. If, however, the objective is to make more general inferences about regeneration or stand dynamics, then modeling random, local multimodality would not be of central interest and the focus should be on the separation of local maxima that reflect the existence and dynamics of subpopulations. For these types of investigations, two-component mixture models are often sufficient [15,18,23,25,30]. While the decision to select a two-, three- or higher-component mixed model may depend on the specific objective of the study, we found that the ungrouped data format gave more similar results among the model families that generally fit the underlying distribution very well. An additional benefit was that no decision on the width of the size classes needed to be made.

It has also been reported that the implementation of mixture models with more than two parameters is problematic, because the estimation process may fail to converge, the algorithm may become extremely unstable, and the global maximum of the likelihood function may not be found [14]. To avoid non-convergence, the GSM model has been promoted as an alternative, because it smooths small local DBH maxima, making it a useful model for approximating the empirical DBH distributions in stratified stands with complex structures [14]. In this study, however, the EM algorithm always led to convergence and robust estimates of parameters. In addition to a consistently superior performance of the two- and three-component log-normal and gamma mixture models over the GSM model, the speed taken by the central processing unit (CPU) for estimating the parameters was much faster for the finite mixture models (i.e., less than 1.5 seconds for all three samples) compared to the time needed for estimating the 250 components of the GSM model (i.e., 120, 150, and 720 seconds for samples 1 and 150 seconds for samples 1, 2, and 3, respectively). and 2, respectively).

The performances of the two- and three-component log-normal and gamma-mixture models were comparable, and often superior, to the approximation obtained using the highly flexible NK model [14,15,38]. This reflects the capacity of two- and three-component log-normal, gamma, and Weibull mixture distributions to very accurately approximate mul-

timodal DBH distributions that describe complex forest structures with three or multiple age cohorts [25]. As exemplified in the two-cohort stands of samples 1 and 2, when DBH distributions reach a first local maximum, decrease thereafter, and increase again to reach a second local maximum (i.e., a rotated sigmoid distribution), two- and three-component mixture models fit very well. In these instances, the mixture models fit around the distinct and sharp local maxima and largely smooth over smaller local maxima, which often express random multimodality and whose overall influence on the quality of the approximation is limited [15]. In contrast, kernel density estimators smooth over the distinct and sharp extremes (i.e., both maxima and empty DBH classes), leading to less precise approximations than in multilayered stands with smoother distributions [15], as was seen with the reverse-*J* DBH distribution in separate heavy-tailed and highly skewed plot whose DBH is not given in this study. The successful performance of the NK model also reflects that an accurate parametric model was selected to fit the empirical DBH data that overcame one of the main limitations that challenge finite mixture distributions, namely the failure of the estimation process to converge or find the global maximum. We conclude that the EM algorithm, at least for the sample plots used in this study, provided initial values that were sufficiently close to the values for which the log likelihood function was able to reach the global maximum.

## 4. Materials and Methods

### 4.1. Material

The EM algorithm was applied to example data collected in three plots located in two different sampling areas. The first sample used two plots of 0.08 ha size that were part of a larger study on the effects of prescribed burning established in multi-aged mixed ponderosa pine (*Pinus ponderosa* Dougl. ex Laws.) that contained scattered western junipers (*Juniperus occidentalis* Hook.). The plots were located in the Malheur National Forest on the southern end of the blue Mountains near Burns, Oregon, USA [39]. Of the many variables originally collected in this study, we only used the diameter at breast height (DBH, measured at 1.3 m height) of all live trees in plot 75 (sample 1) and plot 23 (sample 2). The second sampling area of the study was located in district 1, compartment 111 in the Kordkuy forest in the Golestan province of northern Iran (UTM zone 40: E247508, N4065346). The forest is a semi-natural multi-aged Oriental beech (*Fagus orientalis* Lipsky) dominated forest that is managed with the single-tree selection system. One $100 \times 100$ m plot was established in an area of the compartment that had a tree canopy cover of 85%. All living trees in the plot with a DBH greater than 6 cm were measured using two caliper readings at an angle of $90°$ to one another. The plot summary statistics are provided in Table 6.

**Table 6.** Summary statistics for DBH data.

| Sample | Sampling Area | Number of Trees (n) | Min. | 1st Quartile | Median | Mean | 3rd Quartile | Max. |
|--------|---------------|---------------------|------|--------------|--------|------|--------------|------|
| 1 | USA | 24 | 12.20 | 17.30 | 41.30 | 43.46 | 65.42 | 90.20 |
| 2 | USA | 40 | 11.40 | 24.43 | 31.10 | 31.73 | 34.65 | 71.10 |
| 3 | Iran | 399 | 6.00 | 11.25 | 16.50 | 25.00 | 29.12 | 101.50 |

### 4.2. Methods

#### 4.2.1. Mixture Models and Inference

The PDF of a K-component mixture model has the form

$$g(x|\Theta) = \sum_{k=1}^{K} \omega_k f(x|\boldsymbol{\theta}_k), \tag{1}$$

where $x$ is the DBH value, $\Theta = (\boldsymbol{\omega}', \boldsymbol{\theta}_1', \ldots, \boldsymbol{\theta}_k')'$ in which $\boldsymbol{\theta}_k$ is the parameter vector of the $k$-th component with PDF $f(.|\boldsymbol{\theta}_k)$ and $\boldsymbol{\omega} = (\omega_1, \ldots, \omega_K)'$ is the vector of mixing parameters.

The mixing parameters $\omega_k$s are non-negative and sum to one, i.e., $\sum_{k=1}^{K} \omega_k = 1$. In this study, $f(.|\boldsymbol{\theta}_k)$ denotes the PDF of gamma, log-normal, or Weibull distribution given, respectively, by the following:

$$f(x|\boldsymbol{\theta}_k) = \frac{1}{\beta \Gamma(\alpha)} \left(\frac{x}{\beta}\right)^{\alpha-1} \exp\left\{-\left(\frac{x}{\beta}\right)\right\}, \tag{2}$$

$$f(x|\boldsymbol{\theta}_k) = \frac{1}{x\beta\sqrt{2\pi}} \exp\left\{-\frac{1}{2}\left(\frac{\log(x)-\alpha}{\beta}\right)^2\right\}, \tag{3}$$

$$f(x|\boldsymbol{\theta}_k) = \frac{\alpha}{\beta} \left(\frac{x}{\beta}\right)^{\alpha-1} \exp\left\{-\left(\frac{x}{\beta}\right)^{\alpha}\right\}, \tag{4}$$

where $x > 0$ and $\boldsymbol{\theta}_k = (\alpha, \beta)'$. For the families given in (2) and (4), $\alpha$ and $\beta$ play the role of the shape and scale parameters, respectively. Assuming that $\boldsymbol{x} = (x_1, \ldots, x_n)'$ are coming from a mixture model with PDF (1), the ML estimator $\widehat{\Theta}$ of the parameter vector $\Theta$ is obtained by maximizing the log-likelihood function. This means

$$\widehat{\Theta} = \underset{\Theta}{\operatorname{argmax}} \sum_{i=1}^{n} \log\left(\sum_{k=1}^{K} \omega_k f(x_i|\boldsymbol{\theta}_k)\right).$$

In order to estimate $\widehat{\Theta}$ through the ML approach, computer packages use iterative methods such as the NR algorithm, which does not always converge on the global maximum, particularly for irregular and incomplete distributions with heavy tails. When encountering incomplete data problems, the EM algorithm introduced by [40] is the most popular inferential tool for estimating the parameter vector.

### 4.2.2. Application of Mixture Models to Sample Plots

Mixture models of gamma, log-normal, and Weibull distributions were each fitted to samples 1–3. Groups in all three samples were either 2.5 cm or 5 cm wide (hereafter grouped-2.5 or grouped-5). Parameters of the mixture models in both grouped and ungrouped cases were estimated using the EM algorithm described briefly in Appendices A and B. We note that the initial values for implementing the EM algorithm were obtained through the K-means method of clustering in the R [41] environment using command kmeans(.). To do this, data were partitioned into K groups and the proportion of data points belonging to the $k$-th cluster was considered as the initial values for $k$-th mixing parameter for $k = 1, \ldots, K$. For each group $k$, the initial values for $\alpha_k$ and $\beta_k$ were obtained using the method of moments. For applying the K-means approach to the grouped DBH observations, the midpoint of the $i$-th cluster is repeated $f_i$ times for $i = 1, \ldots, m$ in order to obtain $n$ observations for applying K-means approach. For implementing the EM algorithm, we have provided an R package called ForestFit [42]. Performance of the various mixture models was compared for both grouped and ungrouped data. The performances were further compared with the GSM and NK models using only the ungrouped data. The K-component GSM model with PDF given by

$$g_{gsm}(x|\Theta) = \sum_{k=1}^{K} \omega_k \frac{\beta^k}{\Gamma(k)} x^{k-1} e^{-\beta x},$$

where $\Theta = (\omega_1, \ldots, \omega_K, \beta)'$ has been used recently for modelling the DBH distribution [28]. The NK model estimate the fitted PDF $f(x)$, to the data using

$$f(x) = \frac{1}{nh} \sum_{i=1}^{n} \mathcal{K}\left(\frac{x - x_i}{h}\right),$$

where $x_1, \ldots, x_n$ are DBH observations, $h$ is bin size, and $\mathcal{K}(y)$ is a mathematical function that satisfies $\int_{-\infty}^{\infty} \mathcal{K}(y) = 1$. It should be noted that in the present study, $h$ is determined by the Plug-in bandwidth estimator proposed by [43] and the kernel was the Gaussian type defined as $\mathcal{K}(y) = 1/\sqrt{2\pi} \exp\{-y^2/2\}$. The statistical packages GSM [27] and `kerdiest` [44] that were developed for the R [41] environment were used to fit the GSM and NK models, respectively. Also, for having a fair comparison, we suppose that data are given in ungrouped from. As statistical goodness-of-fit measures, we used the AIC, AD, CVM, KS, and LL given by

$$\text{AIC} = -2\sum_{i=1}^{n} \log g(x_i|\Theta) + 3\text{K} - 1, \tag{5}$$

$$\text{AD} = -n - \sum_{i=1}^{n} \left(\frac{2i-1}{n}\right) \left[\log F(x_{(i)}|\Theta) + \log\left(1 - F(x_{(n-i+1)}|\Theta)\right)\right], \tag{6}$$

$$\text{CVM} = \frac{1}{12n} + \sum_{i=1}^{n} \left[F(x_{(i)}|\Theta) - \frac{2i-1}{2n}\right]^2, \tag{7}$$

$$\text{LL} = \log g(\boldsymbol{x}|\Theta) = \sum_{i=1}^{n} \log g(x_i|\Theta), \tag{8}$$

were $x_{(i)}$ denotes the $i$-th ordered value of DBH and $F(.|\Theta)$ is cumulative distribution function. For computing statistics AIC, AD, CVM, and LL given in (5)–(8), we assume that $\Theta$ is estimated using the EM algorithm. We note that the smaller (larger) values of AIC (LL) indicate better models.

## 5. Conclusions

We derived the expectation-maximization (EM) algorithm for estimating parameters of the most commonly used finite mixture distributions (i.e., gamma, log-normal, and Weibull mixture models) fitted to grouped and ungrouped empirical diameter at breast height (DBH) data. We used three sample plots with different DBH distributions to showcase the EM algorithm, investigated the performance of the mixture models, and compared their performance to nonparametric kernel-based density estimation (NK) and gamma-shaped mixture (GSM) model. We want to stress, however, that while our analysis of three different example sample plots provided very encouraging results for the utility of the EM algorithm to model with irregular diameter distributions, the conclusions about the relative performances of the gamma, log-normal, and Weibull mixture models apply only to these three sample plots and should not be extrapolated beyond these plots. Sample plots represented a bimodal rotated sigmoid distribution without any trees between 30 and 50 cm in size, a bimodal distribution with understory trees up to 45 cm, no trees between 45 and 65 cm in DBH, and a small overstory component between 65 and 75 cm (mixed-age ponderosa pine with scattered western junipers in both plots), and a reverse-*J* diameter distribution with several small modes toward the right tail of the distribution (mixed, deciduous uneven-aged Oriental beech forest). In these mixed-species, two-cohort/two-layered, and multi-cohort/multi-layered stands, the two- and three-component finite log-normal and gamma mixture models modeled empirical DBHs (grouped and ungrouped) very well and generally outperformed the Weibull mixture model. Both of these mixed model types also consistently outperformed the very flexible GSM model that has previously been shown to work well for stands with high skewness and heavy tails. The log-normal and gamma distributions were superior to the NK model in clearly bimodal stands and were inferior in the reverse-*J* situation that contained several minor modes along the right tail of the distribution. Because nonparametric models preclude the computation of statistics such as standard errors and confidence intervals or hypothesis testing of the estimated parameters, parametric mixture models are generally regarded as a more insightful approach for fitting DBH distributions. We conclude that the two- and three-component log-normal and gamma mixture models are well suited to

characterize multimodal DBH distributions for natural stands with two, three or multiple age cohorts of complex structure. The capability of these models to approximate and quantify multimodal empirical DBH distributions makes these models a valuable tool for investigating forest dynamics in complex stands that increasingly guide management approaches in forestry.

**Author Contributions:** Conceptualization, M.T.; methodology, M.T.; software, M.T.; validation, E.K.Z.; formal analysis, M.T.; investigation, E.K.Z., M.T.; original draft preparation, M.T.; review and editing, E.K.Z.; visualization, M.T.; supervision, E.K.Z.; project administration, E.K.Z. All authors have read and agreed to the published version of the manuscript.

**Funding:** This research received no external funding.

**Institutional Review Board Statement:** Not applicable.

**Informed Consent Statement:** Not applicable.

**Data Availability Statement:** Research Data used in this study can be found at https://www.fs.usda.gov/rds/archive/Catalog/RDS-2017-0041.

**Acknowledgments:** Authors would like to thank three anonymous referees for their constructive comments that greatly improved quality of the paper. The second author would like to thank S. Ghalandarayeshi from Department of Statistics of Gonbad Kavous University for sharing experimental data of the sampling area that is located in Iran.

**Conflicts of Interest:** The authors declare no conflict of interest.

**Sample Availability:** Samples of the compounds are available from the second author.

## Appendix A. A Brief Introduction to the EM Algorithm for Mixture Models Fitted to Ungrouped Data

In this study we use the EM algorithm for estimating $\Theta$ in the mixture model (1) when $x$ is available in both grouped and ungrouped (raw) forms. The EM algorithm finds the ML estimators for the parameters of a statistical model when some information about the model is missing. For a $k$-th component mixture model, each observed value is accompanied by a label that determines the observation belongs to which component. For example, the paired sample $(x_i, j)$ shows that $i$-th observed value $x_i$ belongs to $j$-th component (for $i = 1, \ldots, n$ and $k = 1, \ldots, K$) in which $j$ is unknown label or origin of component. Here, we have two sets of data (or variables) including the set of observed variable and latent variable (labels). A sample including both of observed and latent variables is called complete data. For a mixture model corresponds to (1), the observed data (DBH data) are denoted by $x_1, \ldots, x_n$ where the $k$-th component of the mixture has the PDF $f(.|\theta_k)$. In the EM algorithm framework, the observed data corresponding to (1) are denoted by $x_1, \ldots, x_n$ where the $k$-th component of the mixture has the pdf $f(.|\theta_k)$. We denote the complete data by $\xi = (\xi_1', \ldots, \xi_n')' = ((x_1, z_1'), \ldots, (x_n, z_n'))'$ in which $z_i = (z_{i1}, \ldots, z_{iK})'$ is the latent realization of $Z_i = (Z_{i1}, \ldots, Z_{iK})'$ defining the origin component of $x_i$ for $i = 1, \ldots, n$. In each realization of $Z_i$, one of its components, say $Z_{ik}$, equals to 1 and the others are zero. Such realization states that $y_i$ comes from the $k$-th component of the mixture model. We note that by this construction the observation $x_i$ comes from the $k$-th component when $Z_{ik} = 1$, for $i = 1, \ldots, n$ and $k = 1, \ldots, K$. The complete data log-likelihood function, i.e., $l_c(\Theta, \xi)$ is given as

$$l_c(\Theta, \xi) = C + \sum_{i=1}^{n} \sum_{k=1}^{K} z_{ik} \log \omega_k + \sum_{i=1}^{n} \sum_{k=1}^{K} z_{ik} \log f(x_i|\theta_k),$$

where the constant C is independent of parameter vector $\Theta$. Each EM algorithm has two parts, including expectation (E) and maximization (M) steps. Both E- and M-steps are repeated until convergence occurs. In what follows, we introduce both the E- and M-step of the EM algorithm. Assuming that we are at the $(t + 1)$-th iteration of the EM

algorithm, the E-step requires the calculation of the conditional expectation $Q\left(\Theta|\Theta^{(t)}\right)$ of the complete data log-likelihood function given the observed data $x$ and a current estimate $\Theta^{(t)} = \left(\omega'^{(t)}, \theta'^{(t)}\right)'$ of the parameter vector $\Theta$. We have

$$Q\left(\Theta|\Theta^{(t)}\right) = E\left(l_c(\Theta, \xi)|x, \Theta^{(t)}\right) = C + \sum_{i=1}^{n}\sum_{k=1}^{K} \tau_{ik}^{(t)} \log \omega_k + \sum_{i=1}^{n}\sum_{k=1}^{K} \tau_{ik}^{(t)} \log f(x_i|\theta_k), \quad (A1)$$

where

$$\tau_{ik}^{(t)} = E\left(Z_{ik}\big|x_i, \Theta_k^{(t)}\right) = \frac{\omega_k^{(t)} f\left(x_i\big|\theta_k^{(t)}\right)}{\sum_{k=1}^{K} \omega_k^{(t)} f\left(x_i\big|\theta_k^{(t)}\right)}, \quad (A2)$$

for $i = 1, \ldots, n$ and $k = 1, \ldots, K$. The M-step can be carried out for each of PDFs given in (2)–(4) as follows, respectively.

*Appendix A.1. M-Step of the EM Algorithm for Mixture of Gamma Distributions*

Suppose $x = (x_1, \ldots, x_n)'$ denotes a sample of $n$ independent observations from mixture of gamma distributions with pdf given in (2) where $\theta = (\alpha, \beta)'$. Substitute pdf of the gamma distribution into the right-hand side of (A1) to obtain

$$Q(\Theta|\Theta^{(t)}) = C + \sum_{i=1}^{n}\sum_{k=1}^{K} \tau_{ik}^{(t)} \log \omega_k + \sum_{i=1}^{n}\sum_{k=1}^{K} \tau_{ik}^{(t)} \left[-\log \beta_k - \log \Gamma(\alpha_k) + (\alpha_k - 1)\log \frac{x_i}{\beta_k} - \frac{x_i}{\beta_k}\right]. \quad (A3)$$

where C is a constant independent of $\theta_k = (\alpha_k, \beta_k)'$. Assume that we are currently performing the $(t+1)$-th iteration of the EM algorithm. We maximize (A3) with respect to $\alpha_k$ and $\beta_k$ in order to update $\alpha_k^{(t)}$ and $\beta_k^{(t)}$ as $\alpha_k^{(t+1)}$ and $\beta_k^{(t+1)}$, respectively. It follows that $\alpha_k^{(t+1)}$ is a solution of the equation

$$\alpha_k^{(t+1)} = \underset{\alpha_k}{\operatorname{argmax}} \sum_{i=1}^{n} \tau_{ik}^{(t)} \left[-\log \Gamma(\alpha_k) + (\alpha_k - 1)\log \frac{x_i}{\beta_k^{(t)}}\right]$$

and

$$\beta_k^{(t+1)} = \frac{\sum_{i=1}^{n} \tau_{ik}^{(t)} x_i}{\alpha_k^{(t)} \sum_{i=1}^{n} \tau_{ik}^{(t)}}.$$

*Appendix A.2. M-Step of the EM Algorithm for Mixture of Log-Normal Distributions*

Suppose $x = (x_1, \ldots, x_n)'$ denotes a sample of $n$ independent observations from mixture of log-normal distributions with pdf given in (3) where $\theta = (\alpha, \beta)'$. Substitute pdf of the log-normal distribution into the right-hand side of (A1) to obtain

$$Q(\Theta|\Theta^{(t)}) = C + \sum_{i=1}^{n}\sum_{k=1}^{K} \tau_{ik}^{(t)} \log \omega_k + \sum_{i=1}^{n}\sum_{k=1}^{K} \tau_{ik}^{(t)} \left[-\log x_i - \log \beta_k - \frac{1}{2}\left(\frac{\log x_i - \alpha_k}{\beta_k}\right)^2\right], \quad (A4)$$

where C is a constant independent of $\theta_k = (\alpha_k, \beta_k)'$. Assume that we are currently performing the $(t+1)$-th iteration of the EM algorithm. We maximize (A4) with respect to $\alpha_k$ and $\beta_k$ in order to update $\alpha_k^{(t)}$ and $\beta_k^{(t)}$ as $\alpha_k^{(t+1)}$ and $\beta_k^{(t+1)}$, respectively. It follows that

$$\alpha_k^{(t+1)} = \frac{\sum_{i=1}^{n} \tau_{ik}^{(t)} \log x_i}{\sum_{i=1}^{n} \tau_{ik}^{(t)}},$$

and

$$\beta_k^{(t+1)} = \frac{\sum_{i=1}^n \tau_{ik}^{(t)} \log\left(x_i - \alpha_k^{(t)}\right)^2}{\sum_{i=1}^n \tau_{ik}^{(t)}}.$$

*Appendix A.3. M-Step of the EM Algorithm for Mixture of Weibull Distributions*

Suppose $x = (x_1, \ldots, x_n)'$ denotes a sample of $n$ independent observations from mixture of Weibull distributions with pdf given in (4) where $\theta = (\alpha, \beta)'$. Substitute pdf of the gamma distribution into the right-hand side of (A1) to obtain

$$Q(\Theta|\Theta^{(t)}) = C + \sum_{i=1}^n \sum_{k=1}^K \tau_{ik}^{(t)} \log \omega_k + \sum_{i=1}^n \sum_{k=1}^K \tau_{ik}^{(t)} \left[\log \alpha_k - \log \beta_k + (\alpha_k - 1) \log \frac{x_i}{\beta_k} - \left(\frac{x_i}{\beta_k}\right)^{\alpha_k}\right]. \quad \text{(A5)}$$

where C is a constant independent of $\theta_k = (\alpha_k, \beta_k)'$. Assume that we are currently performing the $(t+1)$-th iteration of the EM algorithm. We maximize (A5) with respect to $\alpha_k$ and $\beta_k$ in order to update $\alpha_k^{(t)}$ and $\beta_k^{(t)}$ as $\alpha_k^{(t+1)}$ and $\beta_k^{(t+1)}$, respectively. It follows that $\alpha_k^{(t+1)}$ is a solution of the equation

$$\alpha_k^{(t+1)} = \underset{\alpha_k}{\operatorname{argmax}} \sum_{i=1}^n \tau_{ik}^{(t)} \left[\log \alpha_k + (\alpha_k - 1) \log \frac{x_i}{\beta_k^{(t)}} - \left(\frac{x_i}{\beta_k^{(t)}}\right)^{\alpha_k}\right],$$

and

$$\beta_k^{(t+1)} = \left[\frac{\sum_{i=1}^n \tau_{ik}^{(t)} x_i^{\alpha_k^{(t)}}}{\sum_{i=1}^n \tau_{ik}^{(t)}}\right]^{\frac{1}{\alpha_k^{(t)}}}.$$

For more details about the EM algorithm for mixture models fitted to ungrouped data, we refer reader to [26,45].

## Appendix B. A Brief Introduction to the EM Algorithm for Mixture Models Fitted to Grouped Data

Suppose each element of the random sample $x = (x_1, \ldots, x_n)'$ follows the PDF given in (1). Further assume that sample $x$ has been partitioned into $m$ mutually exclusive groups each of the form $(a_i, b_i)$ for $i = 1, \ldots, m$. We note that the $a_1$ and $b_m$ can be regarded, respectively, as the minimum and maximum observed values and $\cup_{i=1}^m (a_i, b_i) \subseteq \mathcal{S}$ in which $\mathcal{S}$ denotes the support of the distribution. It is worth noting that we just know about the number $n_i$ of observations falling in $(a_i, b_i)$ and the fact that $a_i < x_{ij} < b_i$ for $j = 1, \ldots, n_i$ and $i = 1 \ldots, m$. So, the vector of observed data is given by $y = \left((n_1, a_1, b_1)', (n_2, a_2, b_2)', \ldots, (n_m, a_m, b_m)'\right)'$.

## Appendix C. Initial Values for Implementing the EM Algorithm Applied to Samples 1, 2, and 3

**Table A1.** Initial values of the EM algorithm for sample 1 obtained using the K-means clustering approach when DBH data are ungrouped (UG) and grouped (G) in classes of width 5 cm. It should be noted that the estimated vector of mixing parameters is not given for the sake of saving space.

| | | | Estimated Parameters | |
|---|---|---|---|---|
| K | Type | Family | $\alpha^{(0)} = (\alpha_1^{(0)}, \ldots, \alpha_K^{(0)})'$ | $\beta^{(0)} = (\beta_1^{(0)}, \ldots, \beta_K^{(0)})'$ |
| 2 | G | log-normal | $(2.787, 4.189)'$ | $(0.258, 0.483)'$ |
| | | gamma | $(38.508, 13.458)'$ | $(1.771, 1.356)'$ |
| | | Weibull | $(7.1000, 3.873)'$ | $(72.890, 20.173)'$ |
| | UG | log-normal | $(2.850, 4.190)'$ | $(0.367, 0.264)'$ |
| | | gamma | $(37.567, 11.605)'$ | $(1.820, 1.594)'$ |
| | | Weibull | $(4.807, 2.636)'$ | $(73.300, 20.588)'$ |
| 3 | G | log-normal | $(4.411, 4.120, 2.694)'$ | $(0.277, 0.155, 0.683)'$ |
| | | gamma | $(127.705, 172.082, 13.422)'$ | $(0.621, 0.353, 1.393)'$ |
| | | Weibull | $(35.232, 12.082, 3.499)'$ | $(85.467, 66.073, 20.783)'$ |
| | UG | log-normal | $(0.068, 0.075, 0.367)'$ | $(4.440, 4.138, 2.851)'$ |
| | | gamma | $(0.561, 0.299, 1.820)'$ | $(43.036, 48.327, 37.577)'$ |
| | | Weibull | $(87.668, 65.810, 20.588)'$ | $(7.317, 7.078, 2.635)'$ |

**Table A2.** Initial values of the EM algorithm for sample 2 obtained using the K-means clustering approach when DBH data are ungrouped (UG) and grouped (G) in classes of width 5 cm. It should be noted that the estimated vector of mixing parameters is not given for the sake of saving space.

| | | | Estimated Parameters | |
|---|---|---|---|---|
| K | Type | Family | $\alpha^{(0)} = (\alpha_1^{(0)}, \ldots, \alpha_K^{(0)})'$ | $\beta^{(0)} = (\beta_1^{(0)}, \ldots, \beta_K^{(0)})'$ |
| 2 | G | log-normal | $(4.228, 3.361)'$ | $(0.221, 0.096)'$ |
| | | gamma | $(28.139, 15.983)'$ | $(0.699, 2.291)'$ |
| | | Weibull | $(29.191, 4.347)'$ | $(68.226, 31.488)'$ |
| | UG | log-normal | $(4.257, 3.417)'$ | $(0.209, 0.349)'$ |
| | | gamma | $(484.296, 12.085)'$ | $(0.142, 2.374)'$ |
| | | Weibull | $(10.724, 3.512)'$ | $(70.542, 31.526)'$ |
| 3 | G | log-normal | $(4.241, 2.957, 3.460)'$ | $(0.174, 0.313, 0.322)'$ |
| | | gamma | $(1403.329, 22.372, 86.368)'$ | $(0.048, 0.903, 0.388)'$ |
| | | Weibull | $(47.700, 5.661, 10.943)'$ | $(69.290, 21.868, 35.109)'$ |
| | UG | log-normal | $(4.275, 3.030, 3.449)'$ | $(0.290, 0.257, 0.290)'$ |
| | | gamma | $(484.296, 17.989, 69.850)'$ | $(0.1442, 1.113, 0.470)'$ |
| | | Weibull | $(10.724, 3.356, 7.815)'$ | $(70.542, 21.847, 34.643)'$ |

**Table A3.** Initial values of the EM algorithm for sample 3 obtained using the K-means clustering approach when DBH data are ungrouped (UG) and grouped (G) in classes of width 2.5 cm. It should be noted that the estimated vector of mixing parameters is not given for the sake of saving space.

| | | | Estimated Parameters | |
|---|---|---|---|---|
| **K** | **Type** | **Family** | $\alpha^{(0)} = (\alpha_1^{(0)}, \ldots, \alpha_K^{(0)})'$ | $\beta^{(0)} = (\beta_1^{(0)}, \ldots, \beta_K^{(0)})'$ |
| 1 | G | log-normal | 2.868 | 0.838 |
| | | gamma | 2.544 | 9.841 |
| | | Weibull | 1.217 | 26.718 |
| | UG | log-normal | 2.803 | 0.911 |
| | | gamma | 2.536 | 9.853 |
| | | Weibull | 1.686 | 27.749 |
| 2 | G | log-normal | $(4.076, 2.710)'$ | $(0.349, 0.408)'$ |
| | | gamma | $(14.734, 5.464)'$ | $(4.250, 2.990)'$ |
| | | Weibull | $(4.182, 2.290)'$ | $(68.919, 18.445)'$ |
| | UG | log-normal | $(4.085, 2.660)'$ | $(0.318, 0.511)'$ |
| | | gamma | $(14.740, 5.447)'$ | $(4.246, 2.991)'$ |
| | | Weibull | $(4.096, 2.606)'$ | $(69.054, 18.529)'$ |
| 3 | G | log-normal | $(4.237, 3.330, 2.251)'$ | $(0.171, 0.477, 0.215)'$ |
| | | gamma | $(12.866, 22.071, 17.321)'$ | $(4.797, 1.857, 0.955)'$ |
| | | Weibull | $(5.202, 4.403, 4.124)'$ | $(73.599, 30.952, 13.272)'$ |
| | UG | log-normal | $(4.257, 3.488, 2.561)'$ | $(0.181, 0.342, 0.300)'$ |
| | | gamma | $(30.998, 19.398, 8.810)'$ | $(1.790, 2.315, 1.537)'$ |
| | | Weibull | $(5.726, 4.821, 3.383)'$ | $(77.389, 37.974, 15.182)'$ |
| 4 | G | log-normal | $(4.309, 3.883, 3.125, 2.521)'$ | $(0.295, 0.257, 0.232, 0.435)'$ |
| | | gamma | $(51.062, 35.001, 25.532, 15.781)'$ | $(1.521, 1.343, 0.916, 0.717)'$ |
| | | Weibull | $(8.125, 6.965, 5.706, 4.664)'$ | $(82.462, 50.264, 25.293, 12.383)'$ |
| | UG | log-normal | $(4.310, 3.899, 3.234, 2.463)'$ | $(0.303, 0.077, 0.199, 0.210)'$ |
| | | gamma | $(52.172, 47.226, 28.442, 12.874)'$ | $(1.495, 1.048, 0.911, 0.933)'$ |
| | | Weibull | $(7.217, 7.001, 5.565, 4.124)'$ | $(82.857, 52.662, 28.022, 13.272)'$ |

Fore more details about the EM algorithm for mixture models fitted to grouped data, we refer reader to [46,47].

**Appendix D. Figures A1–A5**

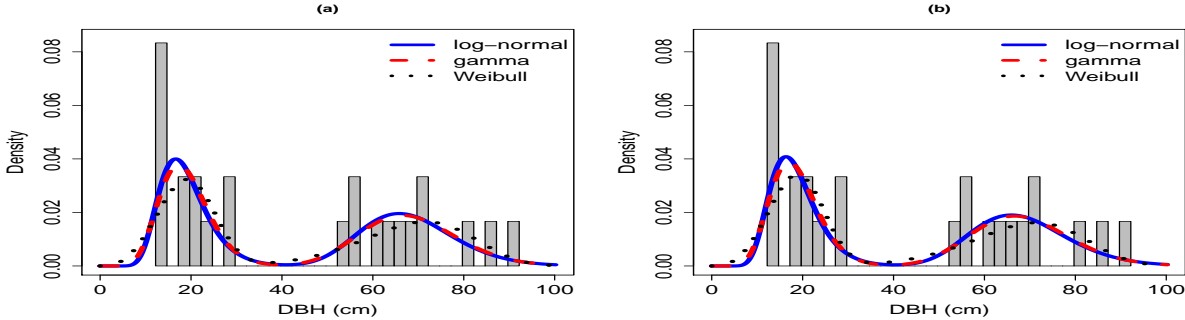

**Figure A1.** *Cont.*

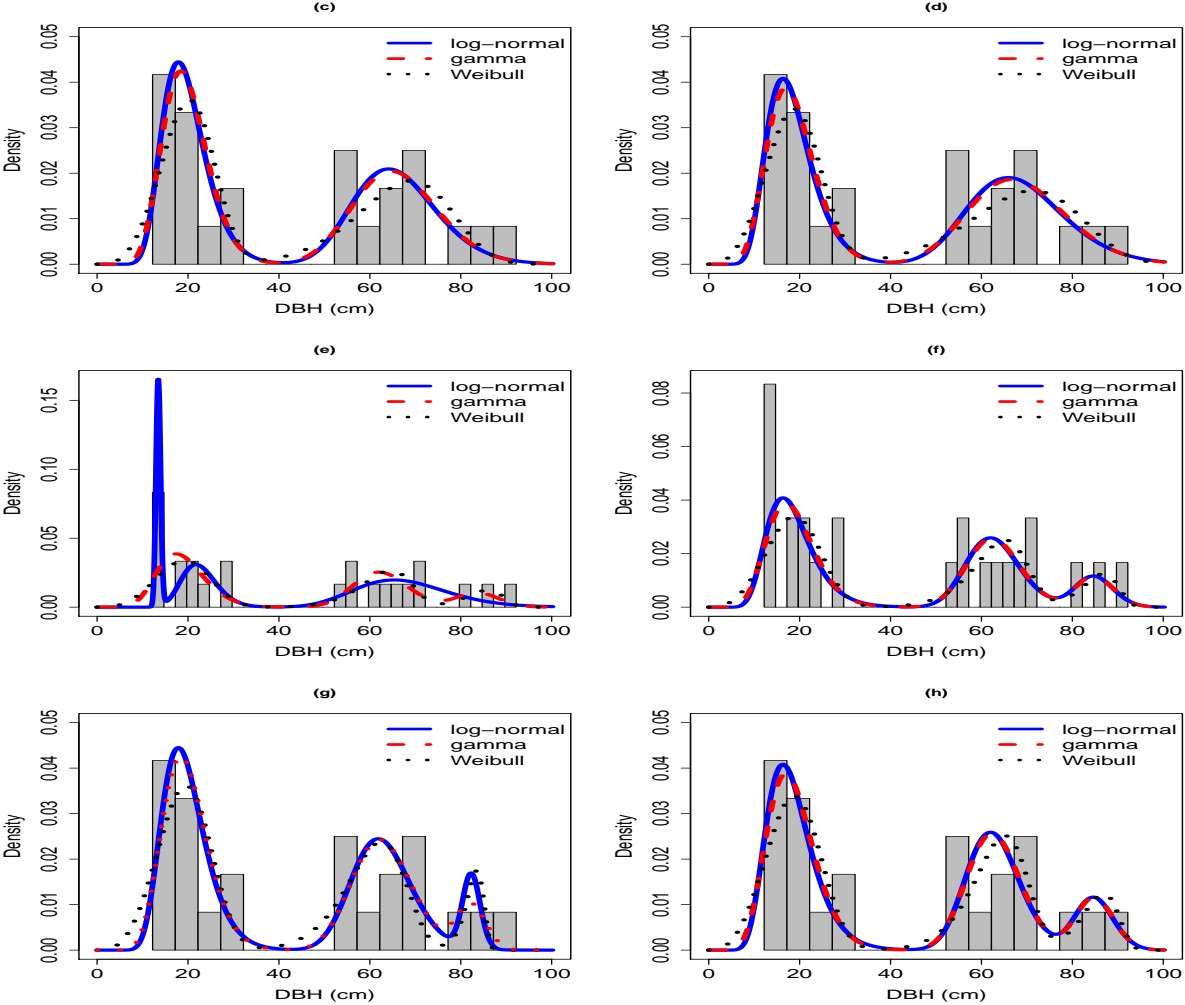

**Figure A1.** Histograms of DBH data in sample 1. (**a**): 2-component mixture models fitted to grouped data corresponding to classes of width 2.5 cm, (**b**): 2-component mixture models fitted to ungrouped data, (**c**): 2-component mixture models fitted to grouped data corresponding to classes of width 5 cm, (**d**): 2-component mixture models fitted to ungrouped data, (**e**): 3-component mixture models fitted to grouped data corresponding to classes of width 2.5 cm, (**f**): 3-component mixture models fitted to ungrouped data, (**g**): 3-component mixture models fitted to grouped data corresponding to classes of width 5 cm, (**h**): 3-component mixture models fitted to ungrouped data. Superimposed are estimated pdf of the mixture of Weibull (solid line), gamma (dashed line), and log-normal (dotted line) distributions.

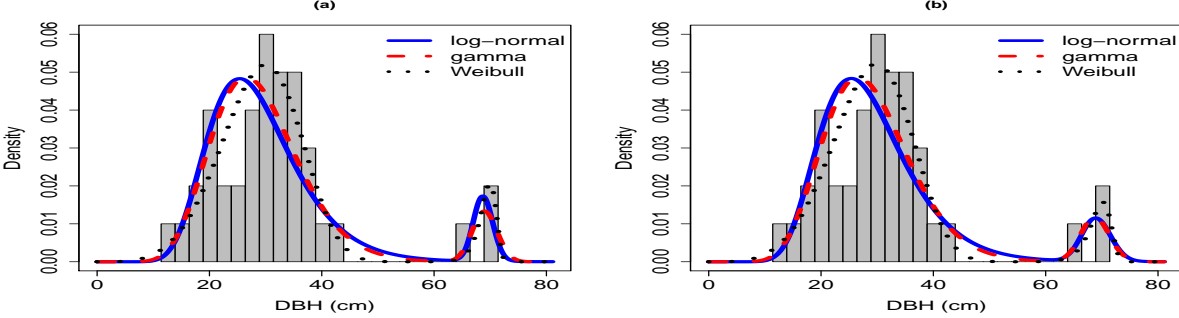

**Figure A2.** *Cont.*

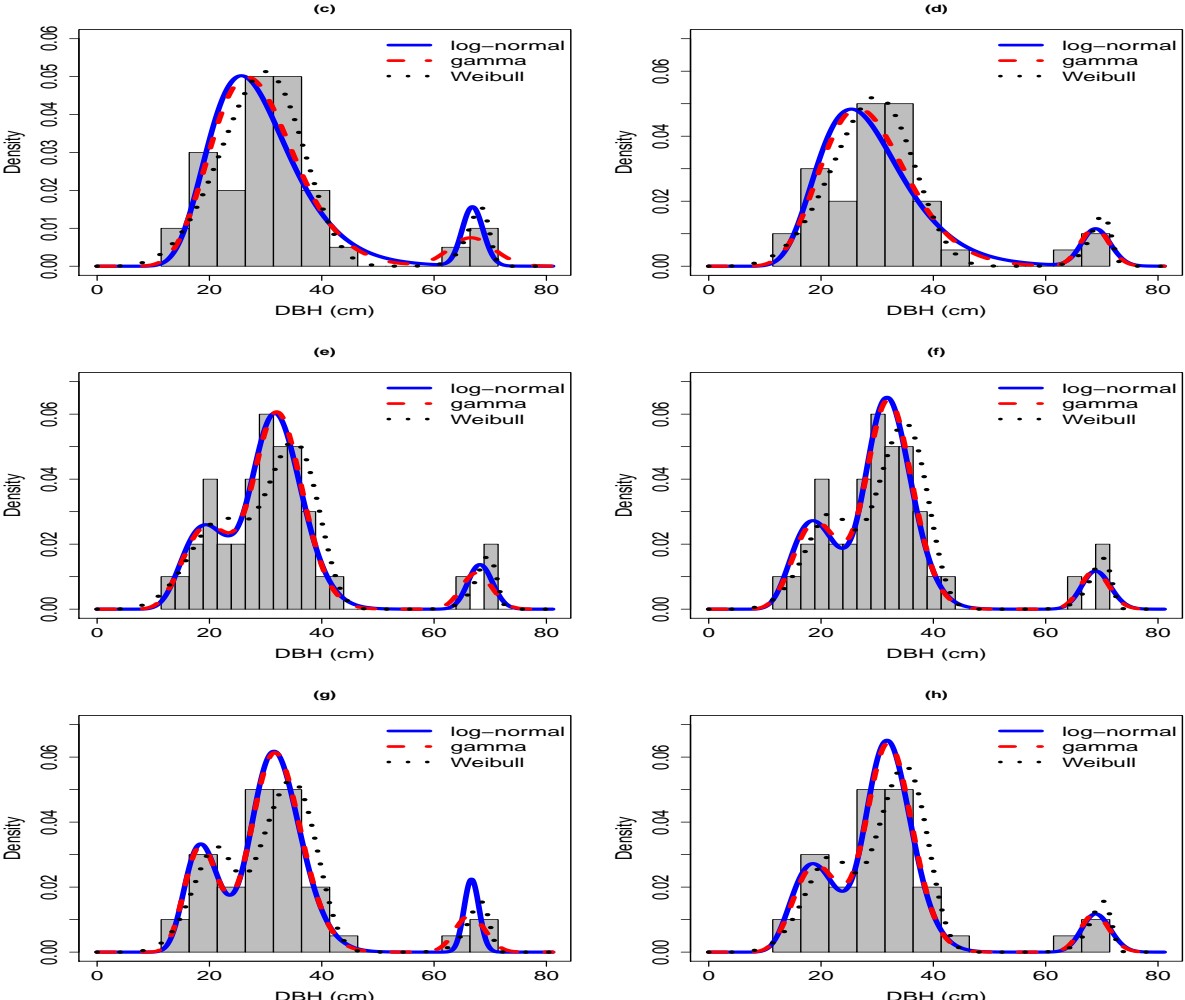

**Figure A2.** Histograms of DBH data in sample 2. (**a**): 2-component mixture models fitted to grouped data corresponding to classes of width 2.5 cm, (**b**): 2-component mixture models fitted to ungrouped data, (**c**): 2-component mixture models fitted to grouped data corresponding to classes of width 5 cm, (**d**): 2-component mixture models fitted to ungrouped data, (**e**): 3-component mixture models fitted to grouped data corresponding to classes of width 2.5 cm, (**f**): 3-component mixture models fitted to ungrouped data, (**g**): 3-component mixture models fitted to grouped data corresponding to classes of width 5 cm, (**h**): 3-component mixture models fitted to ungrouped data. Superimposed in each subfigure are estimated pdf of the mixture of Weibull (solid line), gamma (dashed line), and log-normal (dotted line) distributions.

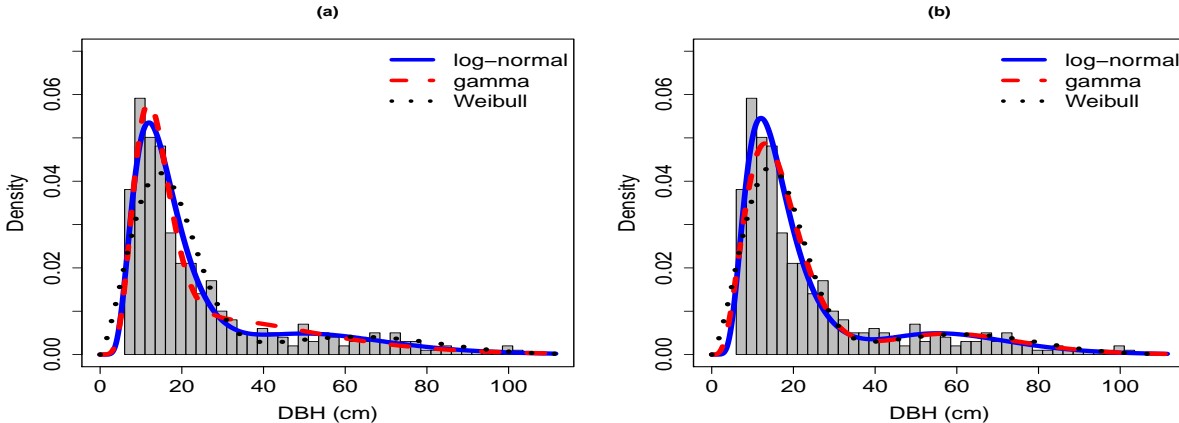

**Figure A3.** *Cont.*

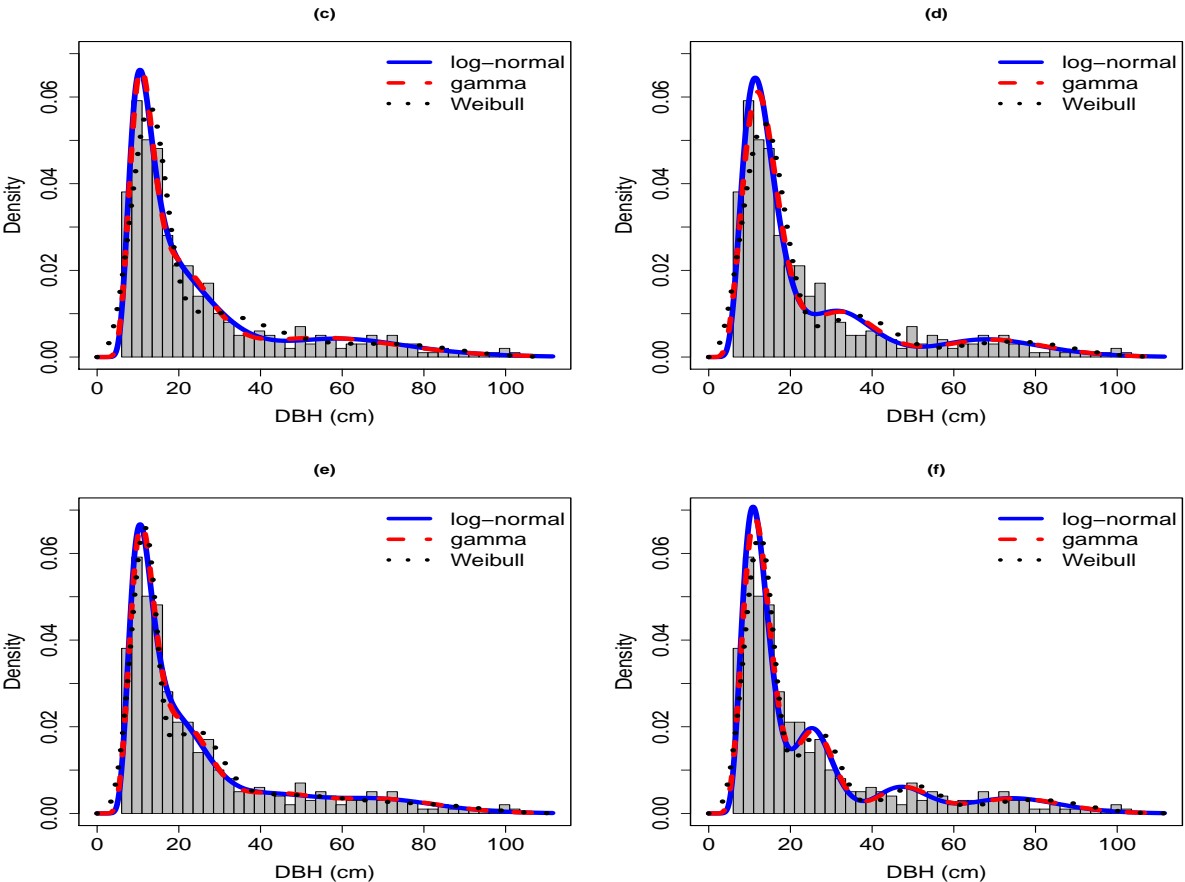

**Figure A3.** Histograms of DBH data in sample 3 corresponding to classes of width 2.5 cm. Superimposed in each subfigure are estimated pdf of the mixture of Weibull (solid line), gamma (dashed line), and log-normal (dotted line) distributions. The fitted pdf in subfigures are related to: (**a**) 2-component mixture models fitted to grouped data, (**b**) 2-component mixture models fitted to ungrouped data, (**c**) 3-component mixture models fitted to grouped data, (**d**) 3-component mixture models fitted to ungrouped data, (**e**) 4-component mixture models fitted to grouped data, and (**f**) 4-component mixture models fitted to ungrouped data.

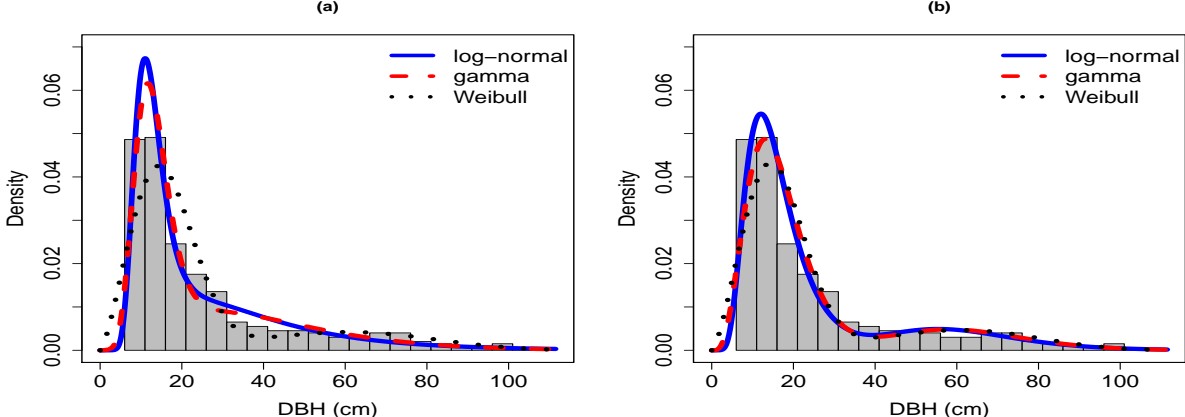

**Figure A4.** *Cont.*

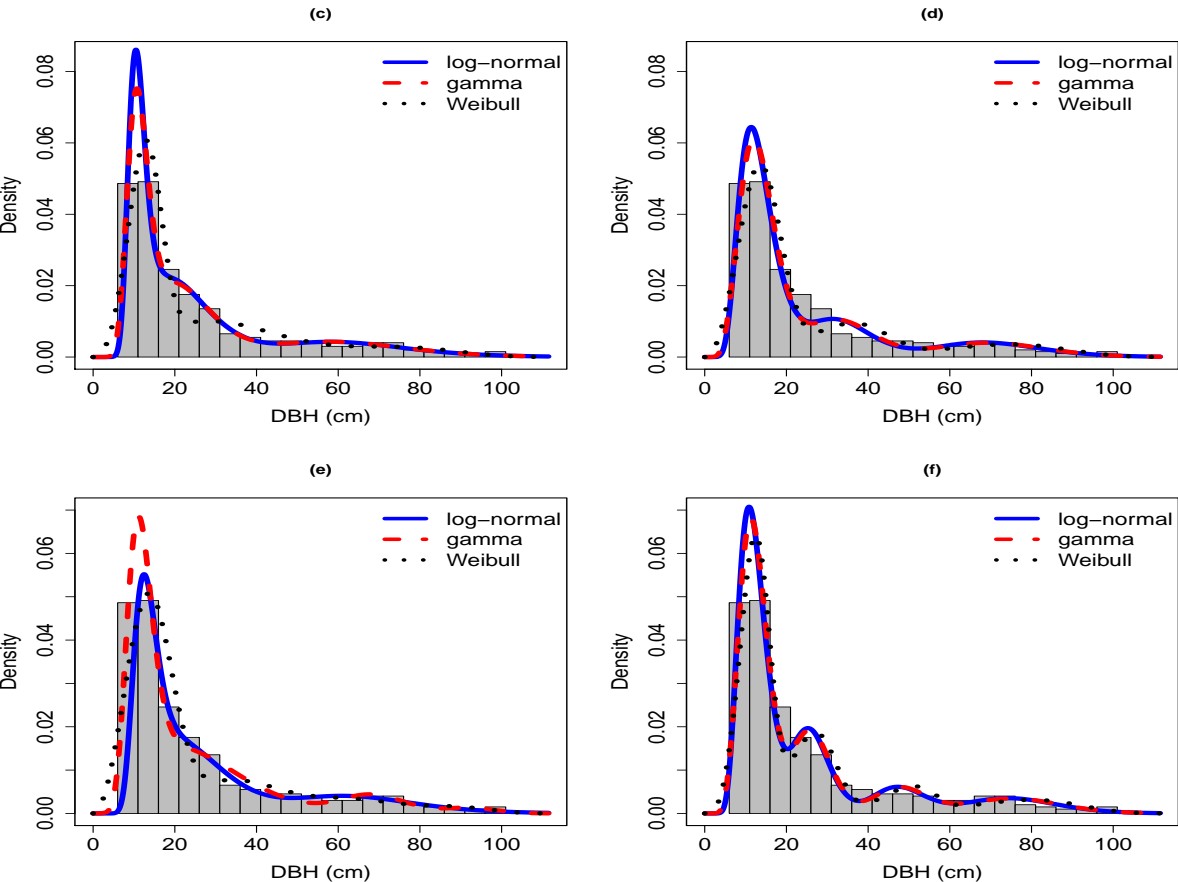

**Figure A4.** Histograms of DBH data in sample 3 corresponding to classes of width 5 cm. Superimposed in each subfigure are estimated pdf of the mixture of Weibull (solid line), gamma (dashed line), and log-normal (dotted line) distributions. The fitted pdf in subfigures are related to: (**a**) 2-component mixture models fitted to grouped data, (**b**) 2-component mixture models fitted to ungrouped data, (**c**) 3-component mixture models fitted to grouped data, (**d**) 3-component mixture models fitted to ungrouped data, (**e**) 4-component mixture models fitted to grouped data, and (**f**) 4-component mixture models fitted to ungrouped data.

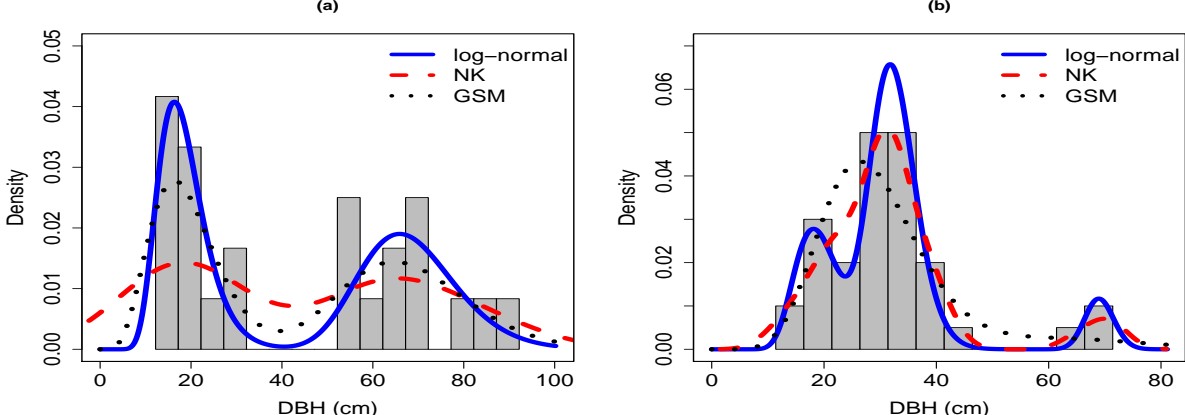

**Figure A5.** *Cont.*

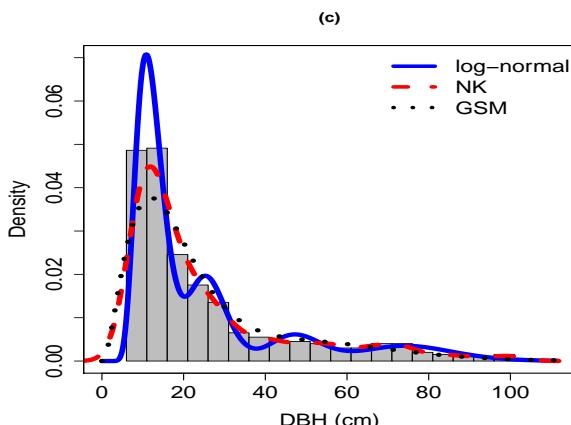

**Figure A5.** Histogram of DBH observations for sample 1 (**a**), sample 2 (**b**), and sample 3 (**c**). Superimposed are pdfs of NK, GSM, and mixture of log-normal models. The pdf of the gamma mixture model has been not shown for the ease of comparison.

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
