# Peer review of "Modeling in Forestry Using Mixture Models Fitted to Grouped and Ungrouped Data"

_forests, doi:10.3390/f12091196_

Round 1

Reviewer 1 Report

A brief summary:

The research refers to the use of a mixture of theoretical functions (Log-normal, Gamma and Weibull) of diameters in order to characterize the structure of stands. The topic is not new but the authors manage to present new aspects in this field.

Broad comments:

Although the topic regarding the analysis of theoretical diameter distributions is well researched so far and the data set used can be considered insufficient (weaknesses), the authors manage by their contribution to highlight new important aspects by which stand structures can be characterized in cases and when there is insufficient field information for investigations.

Specific comments:

Abstract: it is well written, with fluency between sentences.

I noticed that the abbreviations are listed in a separate section, but in my opinion, they should be defined, also in the abstract. In most cases the abstract is the most read (for reasons of access to information or of interest) and by defining abbreviations in the abstract the visibility of research will increase among those who are not knowledgeable in the issue of the article.

The keywords are not found in the Abstract (except dbh distributions and EM algorithm). Please add these keywords in the Abstract or modify the keywords.

INTRODUCTION: This chapter is well presented with information about "state of art" in this field, "and for this purpose, the authors" use a fairly recent and representative bibliography. The aim of this research is clearly defined. I suggest only small changes to be made, which I will present objectively.

L21-23:” Natural disturbances are important drivers of forest dynamics…”

100% is valid for unmanaged forest but for managed forests, humans can influence the dynamics of these ecosystems. Therefore, I consider that this aspect must be specified (in order not to create misunderstandings) in the text by slightly modifying the text, using expressions such as: "in most cases", "in general",… .

L90-102: This part is more about the Materials and methods chapter. I propose to be moved there.

Materials and Methods:

In this chapter the aspects regarding materials and methods are well described and I consider that authors could also make a map with the research area, but this aspect is not mandatory.

L105-17: Does the information used in this research come from uneven-aged stands? This should be specified according to objective research

Results:

The results are presented in a cursive manner and at the end it is observed that the main objective of the research was treated accordingly.

Discussion:

This chapter is in addition to the Results chapter and the research results are well explained and the cited bibliography supports this information.

Table 5 and Table 6 could also introduce the results chapter, but it is not mandatory.

Conclusions:

The conclusions reflect the results of the research.

Appendix:

L301 Appendix C Figures 1-3: I think the graphs would look better if the log-normal function line were more visible

An abnormality in log-normal function may occur in histogram (e) al lower dbh categories. Why do you think it appears?

Author Response

Thank you very much for the constructive comments that improved greatly the manuscript.

Reviewer 2 Report

Review Manuscript # Forests 1342967

The authors in their manuscript attempt to compare the performance of the log-normal, gamma, and Weibull mixture models for complex forest structures. The paper is well written, the objectives are clearly communicated and the analysis is sound. I have only some comments which are currently referred to the following points:

  1. I suggest the authors shortening the length of the abstract according to the journal’s requirements.
  2. Material (L104-110). I suggest clarifying the following points:

-Why only two sample plots were selected?

-Was the sample size sufficient?

-How the selection was made?

  1. L116. I suggest deleting the R (of the two) letter.

Author Response

(The authors gave the same response as above.)

Reviewer 3 Report

Tools to model DBH distributions are required in the frame of forest growth and yield modelling, hence the theme of the work is of interest.
For any kind and type of evaluation I find it quite peculiar to consider only two empirical observations as a benchmarking set.
Moreover I did not find any explanation of the acronyms the work is based on EM, NK and GSM and finally I did not understand how the algorithms you propose work.
If the procedures could be explained, the application to the two available observations could serve as an example application, surely not as an evaluation test.

Author Response

(The authors gave the same response as above.)

Round 2

Reviewer 3 Report

I confirm that the authors have made a nice effort in their revision as all criticized points have been considered. I would personally not be able to make use of their explanations and eventually apply the method they applied to their data. Hence I do not feel sufficiently competent to evaluate if the explanations are "sufficiently improved to warrant publication in Forests". Best regards, Roberto Scotti